# The Alteration of Giglio Island Granite: Relevance to the Conservation of Monumental Architecture

**Fabio Fratini [1],*, Silvia Rescic [1], Oana Adriana Cuzman [1] and Paolo Pierattini [2]**

[1] CNR-Institute of Heritage Science, Via Madonna del Piano 10, Sesto Fiorentino, 50019 Florence, Italy; silvia.rescic@cnr.it (S.R.); oanaadriana.cuzman@cnr.it (Q.A.C.)

[2] Department of Earth Science, University of Florence, Via La Pira 4, 50121 Florence, Italy; paolo.pierattini@unifi.it

\* Correspondence: fabio.fratini@cnr.it

**Abstract:** The research examines the alteration phenomena of Giglio island granite, a rock quarried by Romans from the 3rd century, used for columns in the Italian peninsula and later reemployed in many Christian religious buildings. The study has shown that already in the bedrock there are small percentages of clay minerals. Starting from this condition, the alteration develops by an increase in porosity, which in turn favours the establishment of a slight hydrolysis of the silicates with a decrease in Na, Ca, and K, in accordance with the sericitisation process. The alteration proceeds with a further increase in porosity, apparently not related to a real loss of cohesion, which, however, occurs shortly after, highlighting the necessity of a continuous monitoring of the state of conservation of the material in the architectural heritage.

**Keywords:** granite alteration; Giglio island; granite monuments

## 1. Introduction

Granitic rocks are natural stones that have always fascinated humans, who associated them with characteristics of hardness and great resistance to atmospheric agents. Their use, however, especially because of their hardness, requires special techniques, both for extraction and processing, and has always been linked to technological evolution [1,2].

The examples, even grandiose, of the use of granitic rocks for both structural and decorative elements are widespread all over the world. However, these stones, like all others, are in any case subjected to the decay phenomena that develop in a more striking and rapid manner depending on the textural characteristics and exposure to particular climates that induce transformation phenomena of feldspathic minerals, leading to the complete disintegration of the rock [3–8].

In Italy, the use of granite rocks in monuments and historical architecture is not widespread because there are few outcrops except for the north-eastern part of Sardinia, where these rocks crop out extensively and are still quarried alongside their use as building material in civil architecture [9,10]. For the rest, most of the granite rocks come from the western and central Alps and were used in the 1800s and early 1900s for decorative and architectural elements. Among these, we can mention the granite of Baveno, Montorfano, San Fedelino, Alzo, Quarona, the Traversella diorite, the Balma sienite, and the Adamello tonalite [1,11,12].

However, in the Imperial Age, the Romans made extensive use of granite rocks in the Italian territory, making them arrive above all from the numerous quarries opened in the eastern Egyptian desert and Asia Minor, using them as columns in monumental architecture. In this search for granite rocks, the Romans also sourced to a lesser extent in Sardinia and in two islands of the Tuscan archipelago, Elba and Giglio [1].

As for Giglio island, the granite was quarried particularly from the 3rd century on the east coast of the island. These quarries directly overlook the sea, and this has placed Giglio in a "privileged" condition compared to Elba for the extraction of this rock.

The oldest quarry probably was opened at the time of Julius Caesar and was exploited at least until the early 19th century. It is the *Cava del Foriano,* also called *"alle colonne"* by the islanders, situated above the village of Giglio Porto, with traces of the extraction still visible. Other quarries can be seen near Punta Arenella (*Piccione* and *Arenella* quarries), Punta del Lazzaretto, Scalettino di Giglio Porto, Cala delle Cannelle (*Gran Cava*), and also at Cala delle Caldane (*Bonsere*). The quarries were cultivated intermittently until the middle of the last century (Figure 1).

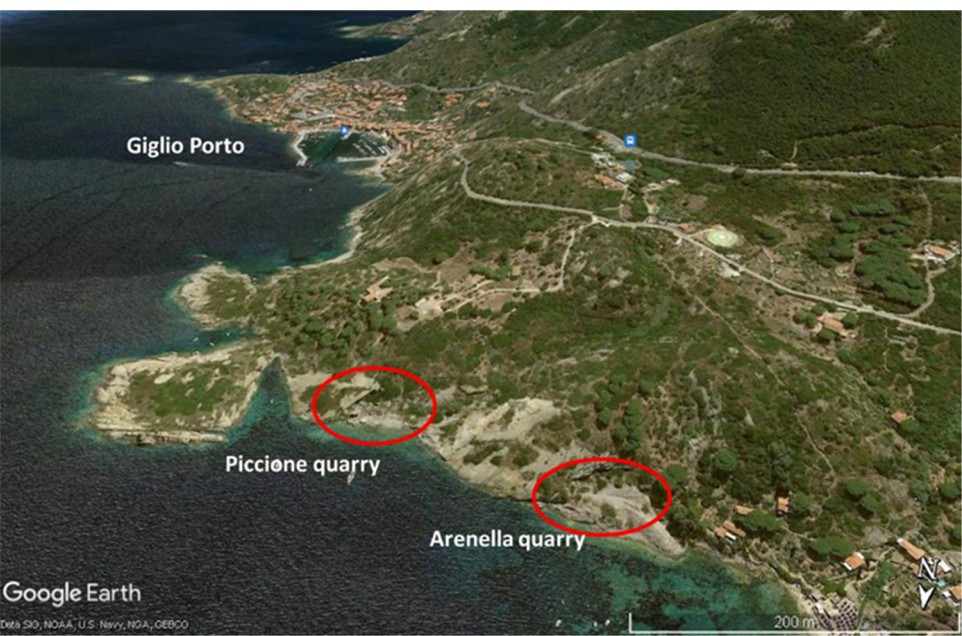

**Figure 1.** Location of the ancient quarries of Piccione and Arenella in the east side of the island (modified from Google Earth).

These quarries were famous above all for the export of beautiful columns, big monoliths weighing several tons, dug directly into the sea front of the quarries, which were then transported by rollers and pontoons on ships. Examples are the six granite columns in the Roman Villa of Domizi Enobarbi in Giannutri island, in the Neronian Port of Anzio, in Rome in the porch of S. Lorenzo in Lucina (first two columns from the right), in the Romanesque porch of via Santi Quattro Coronati, in the pronaos of Santa Croce di Gerusalemme, in the Cancelleria Palace (two columns in the upper and lower colonnade, respectively), in Santa Maria Maggiore (two columns in the left side of the porch), in Sant'Anastasia (four columns, the first and last of the central nave), in Pieve di Santa Maria Assunta di San Leo, in the Forum of Caesar, in Naples in the Church of Gerolomini (12 columns) [12], and in the façade of the Royal Palace (eight columns) from shafts lying in the Foriano quarry. Columns of Giglio origin, probably from the bare of Roman monuments, are present in the Pisa Bell tower (Figure 2), in San Piero a Grado Basilica near Pisa, in the Florence Baptistery, and in the Cathedral of Gaeta. Other columns of later extraction are in the central nave of Pisa Cathedral (four columns in substitution of those damaged by the fire of 1595), in Piazza del Campo in Siena, and in the façades of Marignoli and Bennicelli palaces in Rome.

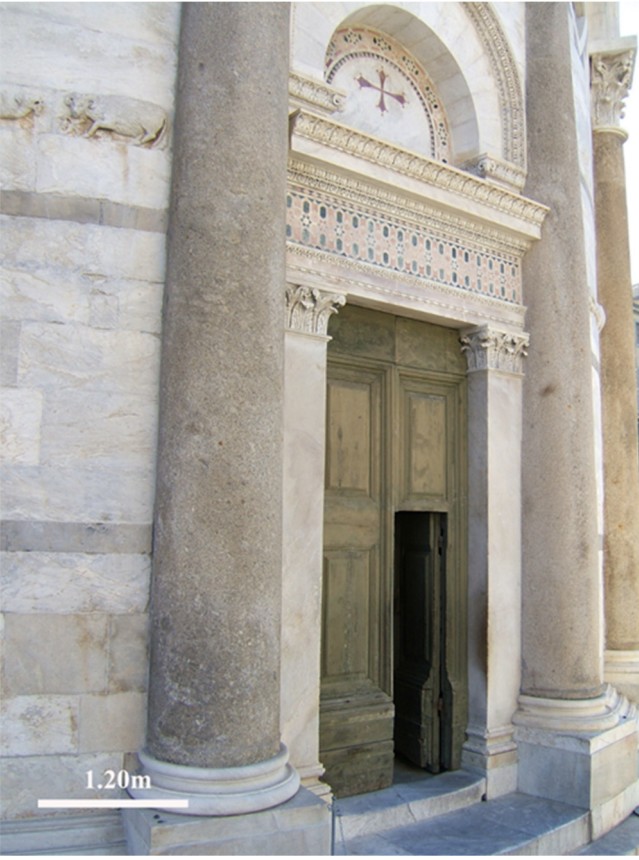

**Figure 2.** Columns of Giglio granite at the base of the Pisa Bell Tower.

On Giglio Island, the granite, used as building material in exposed ashlars, characterises all the civil constructions of the picturesque village of Giglio Castello, which the Medici, in the 16th century, took care to protect from the raids of the pirates by building a mighty circle of walls (Figure 3a). Currently 7 of the 10 original towers and the suggestive three-door entrance remain (Figure 3b).

Observing the whole complex, a brown colour is evident, which is due to the intense chromatic alteration of the rock and constitutes one of the first phases of the decay, which is macroscopically manifested by intercrystalline decohesion, a phenomenon common to all granite rocks (Figure 4a,b).

Moreover, extended areas colonised by biological formations can be observed on the granitic surfaces, their development being influenced by the weathering phenomena with an increasing of the stone bioreceptivity [13] (Figure 4c).

In this research, the alteration processes that take place in this type of material exposed to the exogenous environment have been studied to contribute to the conservation of an architectural heritage, which is limited but of great interest due to its uniqueness in the Italian landscape.

For this purpose, quarry materials with a different state of alteration were sampled, starting from the unweathered rock. The petrophysical characteristics such as total open porosity and saturation index, as well as the mineralogical and chemical characteristics, were evaluated with particular attention to the presence of minerals originating from the alteration of the original minerals (formation of clayey minerals).

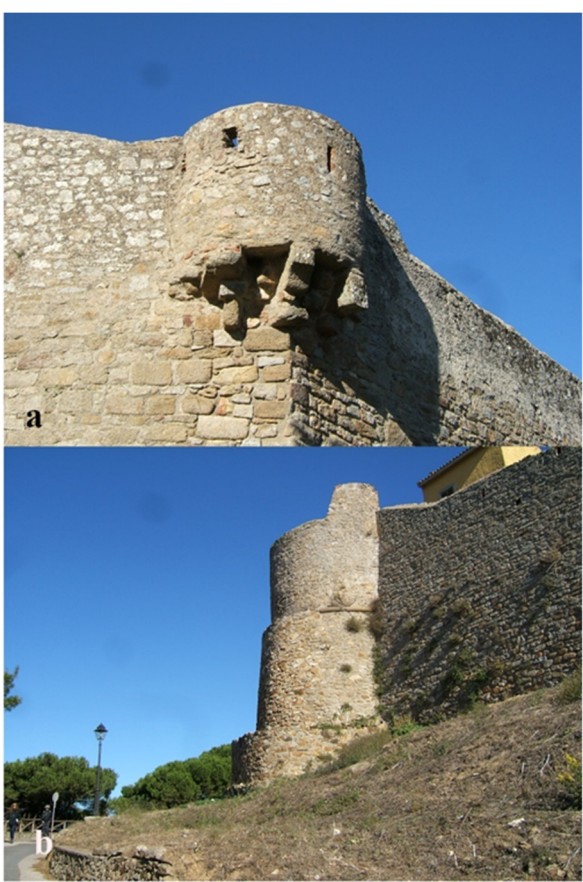

**Figure 3.** Giglio Castello: walls of the 16th century, built with granite ashlars (a). Giglio Castello: detail of a tower of the city walls resting on a granite outcrop (b).

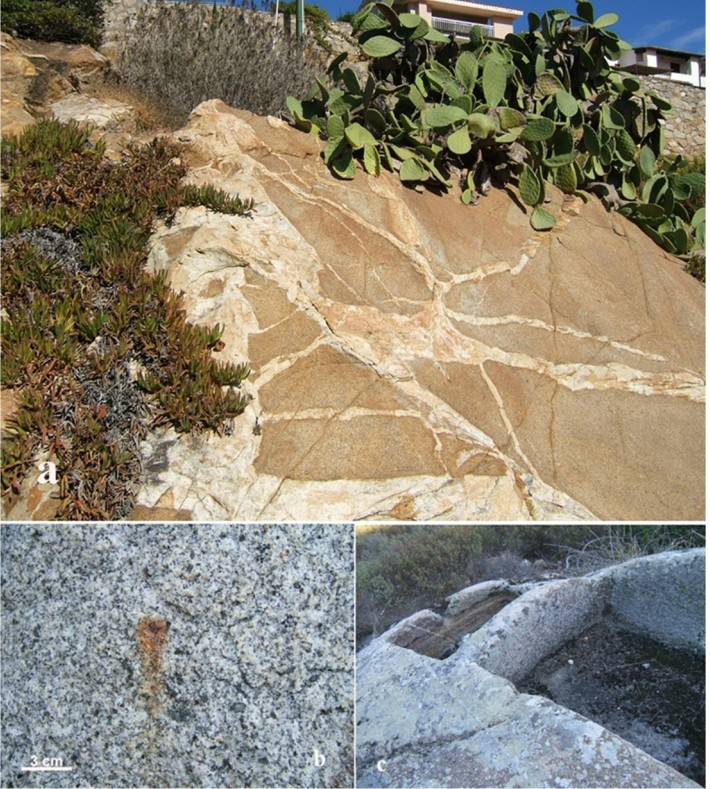

**Figure 4.** Chromatic alteration of the original grey colour due to oxidation phenomena in a granite outcrop at the base of Giglio Castello city walls. The white veins are the filling by quartz (a);

oxidation of a pyrite crystal (b); bio-colonisation in tanks of a *palmento,* a traditional construction for the processing of grape must (c).

## 2. Geological Setting

Giglio Island belongs to the Tuscan Archipelago and is located in the stretch of sea facing the Argentario promontory. The 90% of its extension is made of a monzogranite intrusion that rises to the surface and is linked to the extensive tectonic phase following the collision between the Adria and Corsica–Sardinia plates that formed the Apennine chain. In this tectonic domain, a series of intrusive, volcanic, and/or sub-volcanic magmatism episodes, known in literature as the Tuscan Magmatic Province, took place [14,15].

In the area occupied by the Tuscan Archipelago, starting about 7 Ma ago, four granodiorite and quartz–monzonite intrusive bodies have been emplaced, two in correspondence with Elba Island (Monte Capanne and the Porto Azzurro–Rio Marina area), one in Montecristo, and another in Giglio. This activity developed from the end of the Miocene to the whole Pliocene (from 7 to around 2 Ma ago) [16–18].

This magmatism is generally interpreted as deriving from a large-scale melting process of clayey sediments inside the Earth's crust (anatexis phenomenon). The intrusion of the monzogranite mass took place inside sedimentary rocks, whose remnants from contact metamorphism can be observed at Punta del Fenaio (north of the island) in two small outcrops of strongly foliated phyllites. Some authors report other small outcrops of intensely metamorphosed schists injected with aplitic and quartz veins in the locality of "Vena".

The only part of the island not made up of plutonites is that of the Franco promontory, south-west of Campese village, which also diversifies morphologically. In this area, metamorphic and sedimentary Mesozoic rocks crop out, referable to two distinct structural units, the upper unit, consisting of grey argillites and metagabbros, and the lower unit consisting of crystalline limestone, cavernous limestone, quartz conglomerate, and quartzites of the Verrucano formation.

The intrusions of these plutons caused the lifting of the sedimentary cover and the formation of a horst in correspondence with the Argentario itself. Subsequently, the evolution of the extensional tectonics caused a sinking of the area between Argentario and Giglio, transforming the latter into an island.

Monzogranites are acidic intrusive rocks characterised by a mineralogical composition in which quartz (20%), K-feldspar (20%), and plagioclase (40%) predominate [19]. Other minerals are represented by biotite (10%) and cordierite and as the accessory minerals apatite, zircon, titanite, iron oxides, and sulphides. The minerals' grain sizes range from medium to fine (one to four millimetres), sometimes with centimetre-sized phenocrysts. The rock has a classic whitish grey colour that changes to rusty brown upon weathering.

Two distinct magmatic intrusions [18], that of the Giglio monzogranite intrusion (GMI) occupying the most part of the island and that of the limited outcrop of *Le Scole* islets, south-east of Giglio Porto, known as Scole monzogranite intrusion (SMI), are recognised.

Within the GMI intrusion, two facies can be further distinguished, called "facies of Pietrabona" (PBF) and "facies of Arenella" (ARF), characterised by some petrographic and structural differences. The Pietrabona facies is characterised by very pronounced lineation given by the preferential orientation of flat or elongated micro-granular enclaves, while the Arenella facies is characterised by a more massive texture (only localised orientation of the megacrysts), abundance of K-feldspar megacrysts, and a lower amount of biotite. A further macroscopic characteristic of ARF is the presence of auto-intrusions of micro-granular enclaves with different contents of megacrysts.

The monzogranite of the Le Scole facies (SMI) has a more acidic chemical composition than GMI and is practically free of xenoliths.

The monzogranites undergo an alteration due to two phenomena often connected to each other. The first is essentially linked to fracturing, which creates practically isolated blocks. The second is due to alteration processes (kaolinisation of the feldspars), causing a disruption of the rock, which transforms into a kind of sandstone with a predominantly quartz composition (regolith), which allows for the growth of vegetation. Because of these processes, in some areas, debris accumulations are formed that are easily mobilised by gravitational phenomena and/or by the action of rainwater.

## 3. Materials and Methods

### 3.1. Sampling and Macroscopic Description

In this study, the Arenella facies was examined as it is the one that has been exploited and used also outside the island. This facies shows the following characteristics:

- fresh cut grey colour;
- essentially massive structure with absence of foliation;
- medium grained;
- relatively rich in K-feldspar megacrysts.

Where the alteration is more intense, the granite has the typical reddish-brown colour of the altered acidic granite rocks and becomes extremely friable until it becomes first a sand, then a soil (Figure 5).

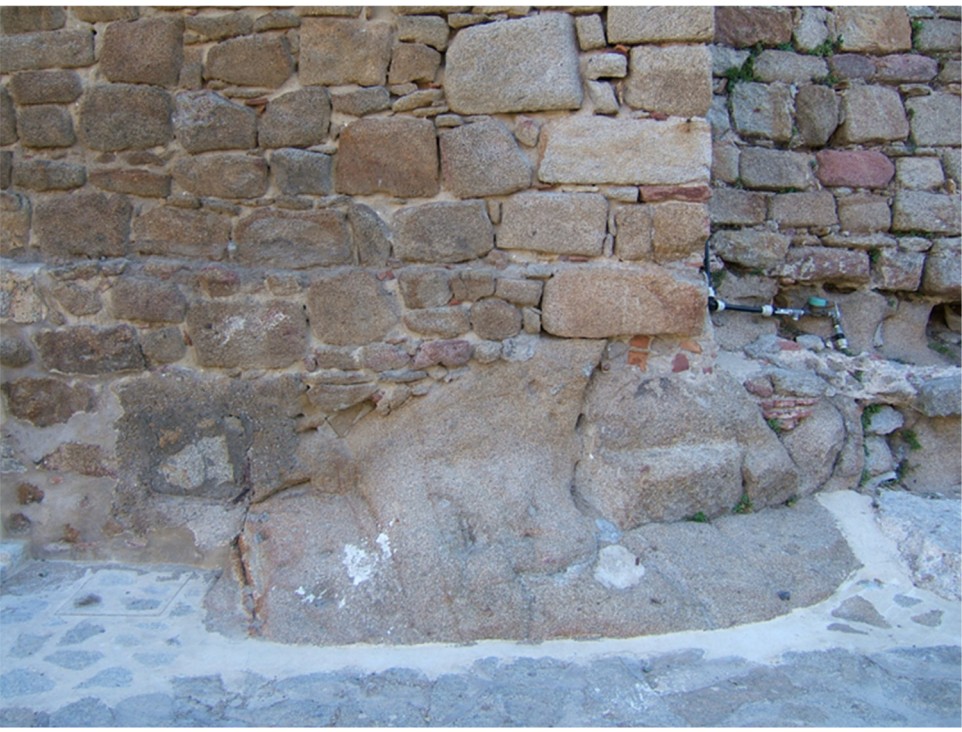

**Figure 5.** Detail of a granite ashlar masonry in which the intense chromatic alteration is evident (the rock changes from an overall grey colour to a "rust" colour) and the strong intergranular disintegration.

Nine vertical alteration profiles distributed along the roads of the island were examined (Figure 6).

The types sampled within each profile, in order of increasing alteration, are the following:

type A—unweathered rock;
type Aa—compact rock with some fractures and slightly ochre colour;

type B—friable rock of ochre colour. The apparent structure is similar to the compact rock of type Aa, but the sample can be broken by hand with a simple pressure;

type C—rock with loss of coherence;

type Cc—soil that generally contains organic material.

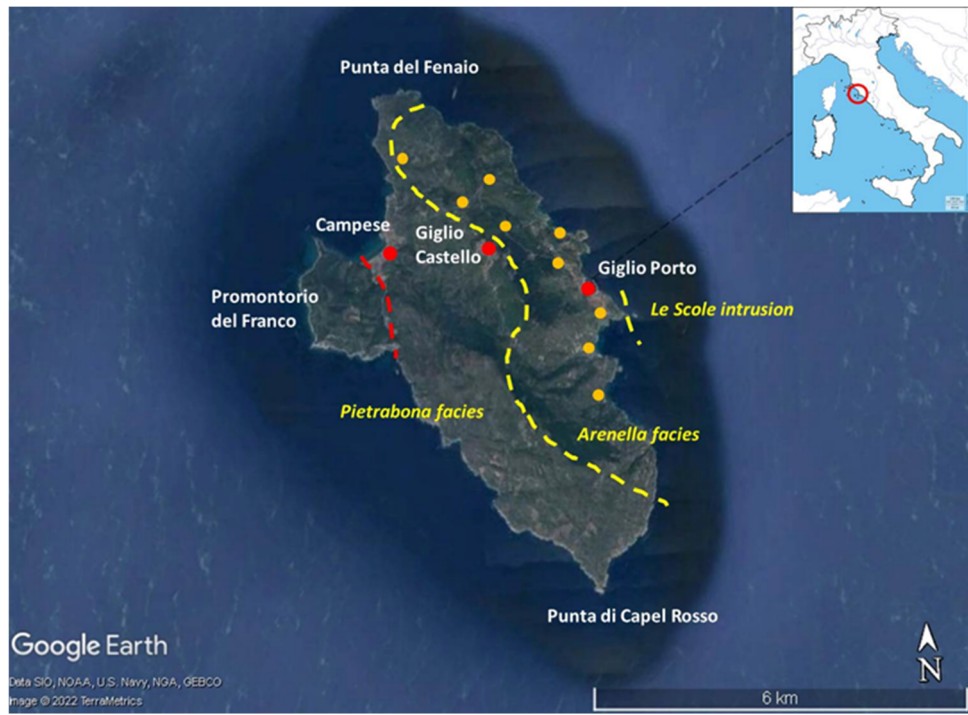

**Figure 6.** Detail of the nine sampling points along the roads of the island (yellow dots). The two yellow dashed lines separate, individually, the Pietrabona facies from the Arenella facies and the latter from the Scole Monzogranite Intrusion. The red dashed line separates the Franco promontory characterised by metamorthic and carbonate Mesozoic rocks (modified from Google Earth).

*3.2. Analytical Methods*

The following investigations were conducted:

- Petrographic study in the thin section with a polarising ZEISS Axioscope A1 microscope (on all types except for the Cc because it is inconsistent). Anorthite (An) content in plagioclase was determined under microscope according to the Michel–Lévy method [20]. The petrographic classification of the rock was carried out by modal analysis on the 9 samples of type A, from three orthogonal thin sections for each sample, according to the Streckeisen's classification [21].

- Determination of clay mineral composition by X-ray diffraction according to Cipriani's methods [22] (on all samples), utilising a PHILIPS PW 1729 diffractometer operating at 40 KV, 20 mA with a CuK $\alpha$1= 1,545Å radiation, a scanning speed= 2∘/min in a range of 2θ = 3-20°. The diffractometer analysis was performed on the <4 μm clay fraction of the bulk samples, previously crushed and powdered, extracted after washing and settling according to the Stokes law. The slide of the <4 μm clay fraction was analysed untreated, after a treatment with ethylene glycol, and after heating at 450 ∘C and 600 ∘C.

- Determination of the chemical composition of conventional major and minor elements by X-ray fluorescence spectrometry (on all samples) on pressed powder pellets, adopting a Philips PW 1480 wavelength-dispersive spectrometer; major and minor elements were determined using a Rh anode and corrected for the matrix effect according to the method of Franzini et al. [23]. Error was evaluated to be less than 1% for the major elements and 5% for the minor elements. LOI (loss on ignition) was determined

by measuring the mass loss in the sample powders heated at 950 °C for 1 hour. FeO was determined by volumetric titration according to the method of Shapiro and Brannock [24]. The chemical compositional was expressed as oxide weight (%) normalised to 100%.

- Determination of the following physical parameters: total open porosity (P%), imbibition coefficient (CIv%), and saturation index (IS%) [25]. The equipment used to determine these parameters were a Quanta Chrome helium pycnometer, a Chandler Engineering mercury pycnometer, and analytical balance. CIv % was determined by total immersion of the specimens in water [26]. These physical investigations were carried out on types A, Aa, and B on 5 specimens for each sample, and the average is reported in the results. On the C and Cc types, it was not possible to carry out such investigations due to the inconsistency of the material.

- The most common phototrophic colonisers and their effect on the granite were mainly observed in situ by using different magnification lenses, and only the black patina alteration was sampled for microscopic analysis, the identification being made according to the specific morphological characteristics of different phototrophic organisms [27,28].

## 4. Results and Discussion

*4.1. Petrographic Study*

The rock has a heterogranular and hypidiomorphic texture with crystal size variable from 1 to 2 mm for plagioclase and biotite, and up to 4-5 mm for K-feldspar.

The principal minerals are the following:

*Plagioclase*—normally twinned according to the albite law, it has euhedral, zoned crystals, with a decrease in the anorthite content from the core towards the rim. The most frequent terms are andesine-oligoclase.

*K-feldspar*—this mineral is always anhedral in large crystals, and perthitic unmixing is often present.

*Quartz*—it occurs in anhedral crystals of considerable size, often showing a recrystallisation into smaller crystals whose formation is linked to dynamic stress phenomena.

*Biotite*—medium-sized, brown, and pleochroic euhedral crystals. It often contains accessory minerals and oxides.

The accessory minerals are the following: tourmaline, cordierite always altered in pinitic products, apatite, monazite, zircon, and oxides.

The modal analysis of the type A (unweathered granite) made it possible to classify the rock as a monzogranite, according to the Streckeisen's classification [21].

In the sections of types Aa, B, and C there is a progressive increase in species linked to the alteration (as for the thin sections of type A, Aa, B, and C, see Supplementary Material 1). In particular, the notable increase in sericite at the expense of feldspars is observed, namely, plagioclase starting from the more anorthitic core of the crystals (Figure 7a). Biotite often shows discoloration phenomena due to hydrothermal alteration with transformation in chlorite (Figure 7b). Chloritisation is irregularly distributed. Indeed, chlorite first develops as a pseudomorph on the primary biotite by interaction of hydrothermal fluids circulating in the fractures of the pluton in the final phase of cooling and setting.

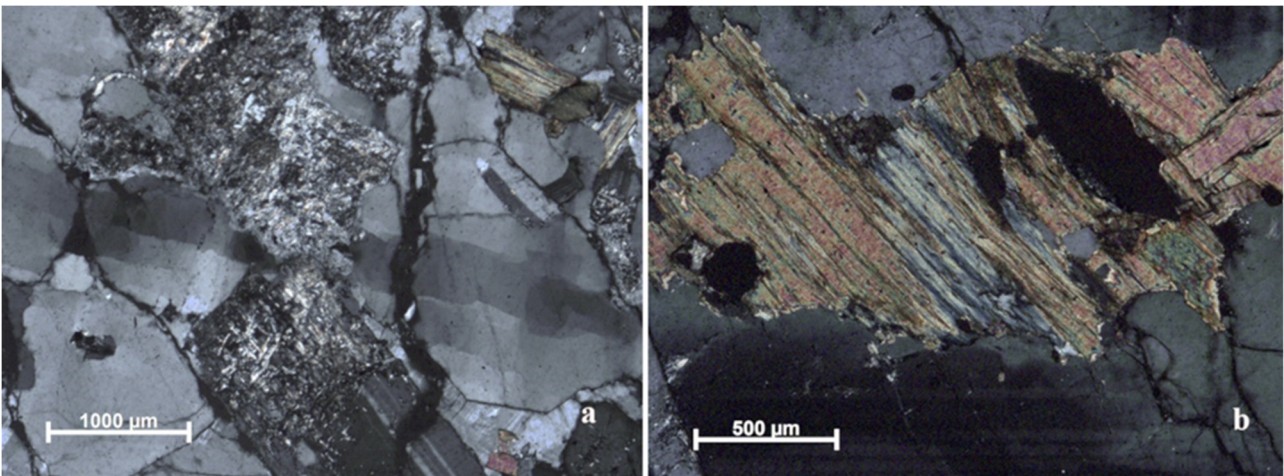

**Figure 7.** Image in petrographic thin section (crossed polarisers): in the centre of the image there is a plagioclase affected by sericitisation phenomena (sericite with higher interference colours (yellowish) (a); image in petrographic thin section (crossed polarisers). In the centre of the image, there is a biotite with partial transformation in chlorite (b).

Another species of probable secondary origin is muscovite linked to the alteration of K-feldspar. The presence of well-formed muscovite crystals in aggregates may also be referred to as a possible primary origin for this mineral.

The observation of types B and C in the thin section also highlights both inter- and intracrystalline fracturing; these fractures can be filled with titanite, epidote, and Fe oxides.

### 4.2. Study of the Mineralogical Composition of the Clay Fraction

The investigation on the clay minerals (Table 1) was carried out with the aim of highlighting the possible presence of alteration minerals of the main constituents, especially feldspar and biotite (as for the XRD diagrams of clay minerals, see Supplementary Material 2). Actually, the analysis of the unweathered samples has shown the constant presence of kaolinite and chlorite, probably due to hydrothermal alteration phenomena. Kaolinite, with increasing alteration, tends to strongly decrease and then reappears in type C. Chlorite, on the other hand, has a more irregular trend. In types B and C, there are also interstratified minerals such as chlorite-vermiculite and illite-smectite, which have their maximum concentration in types B and C, while they drastically decrease in type Cc.

**Table 1.** Mineralogical composition of clay fraction.

|       | Cl   | K    | ClV | ISm |
|-------|------|------|-----|-----|
| **A** | ++++ | ++++ | -   | -   |
| **Aa**| +    | +    | -   | -   |
| **B** | +    | +    | ++  | +++ |
| **C** | +    | ++   | +++ | ++  |
| **Cc**| ++   | +++  | +   | +   |

Cl = chlorite; K = kaolinite; ClV = chlorite-vermiculite; ISm = illite-smectite. ++++ very abundant; +++ abundant; ++ medium; + scarce, - not detected.

The behaviour of kaolinite could be due to an instability of this mineral, which in the early stages of the exogenous alteration turns into illite when it comes into contact with solutions rich in potassium for hydrolysis processes of the feldspar of the horizons above. Its reappearance in the most altered types must be attributed to the process of hydrolysis with leaching, in particular of Na and K, as can be seen in the data obtained from the chemical analysis that will be discussed in the next paragraph.

The interstratified minerals of the most altered samples (types B and C) should be related above all to the alteration of biotite.

### 4.3. Geochemical data

Table 2 shows the average values of the chemical analysis of the major elements relating to the unweathered granite (quarry sample and type A) and to the types with increasing alteration (Aa, B, C, Cc). Data relating to unweathered granite were used to classify the rock according to the method of Streickeisen–Le Maitre [29].

**Table 2.** Chemical composition of major and minor elements from unweathered (A) to weathered samples (Aa, B, C, Cc) expressed as oxide weight %.

|  |  | $SiO_2$ | $TiO_2$ | $Al_2O_3$ | $Fe_2O_3$ | FeO * | MnO | MgO | CaO | $Na_2O$ | $K_2O$ | $P_2O_5$ | LOI ** |
|---|---|---|---|---|---|---|---|---|---|---|---|---|---|
| **A** | M | 68.44 | 0.63 | 15.27 | 0.69 | 2.79 | 0.06 | 1.29 | 2.21 | 2.69 | 4.73 | 0.18 | 1.02 |
|  | σ | *1.46* | *0.08* | *0.45* | *0.17* | *0.35* | *0.01* | *0.25* | *0.34* | *0.14* | *0.32* | *0.03* | *0.28* |
| **Aa** | M | 68.99 | 0.59 | 15.28 | 1.15 | 2.23 | 0.06 | 1.09 | 1.88 | 2.50 | 4.88 | 0.17 | 1.12 |
|  | σ | *1.09* | *0.11* | *0.29* | *0.35* | *0.31* | *0.01* | *0.28* | *0.34* | *0.20* | *0.29* | *0.02* | *0.20* |
| **B** | M | 68.22 | 0.64 | 15.79 | 1.85 | 1.85 | 0.06 | 1.11 | 1.61 | 2.46 | 4.67 | 0.19 | 1.60 |
|  | σ | *0.79* | *0.09* | *0.45* | *0. 8* | *0.75* | *0.01* | *0.21* | *0.22* | *0.14* | *0.27* | *0.02* | *0.35* |
| **C** | M | 67.25 | 0.65 | 16.16 | 2.80 | 1.30 | 0.05 | 1.11 | 1.36 | 2.27 | 4.42 | 0.17 | 2.46 |
|  | σ | *1.06* | *0.08* | *0.51* | *0.90* | *0.79* | *0.01* | *0.18* | *0.30* | *0.12* | *0.47* | *0.03* | *0.62* |
| **Cc** | M | 64.86 | 0.68 | 17.25 | 3.19 | 1.10 | 0.06 | 1.05 | 1.26 | 2.14 | 4.42 | 0.16 | 3.82 |
|  | σ | *2.15* | *0.11* | *0.84* | *1.13* | *0.40* | *0.01* | *0.11* | *0.11* | *0.22* | *0.27* | *0.03* | *1.05* |

M = average of 9 samples; σ = standard deviation; * volumetric titration method; ** loss of ignition.

The results confirmed the data of the petrographic classification by modal analysis, namely, the belonging of the rock to the monzogranite composition [30]. Moreover, the alumina saturation index (ASI = $Al_2O_3$/(CaO+$Na_2O$+$K_2O$) was greater than 1, as expected in peraluminous rocks (A=1.1;) [31]. These data make it possible to assign these monzogranites to type S according to the classification of Chappel and White [31], i.e., origin by fusion of metasedimentary rocks distinguishing them from type I, originated by melting of igneous rocks.

The behaviour of the major elements with increasing alteration is also shown in the diagram of Figure 8a, where the variation in concentration of the elements should be correlated with the alteration of the minerals present in the rock and with the conditions in which it develops.

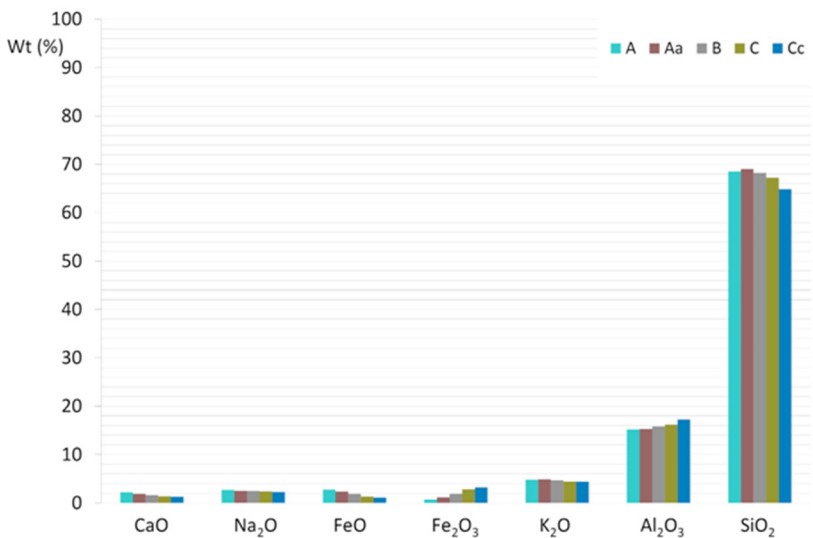

a

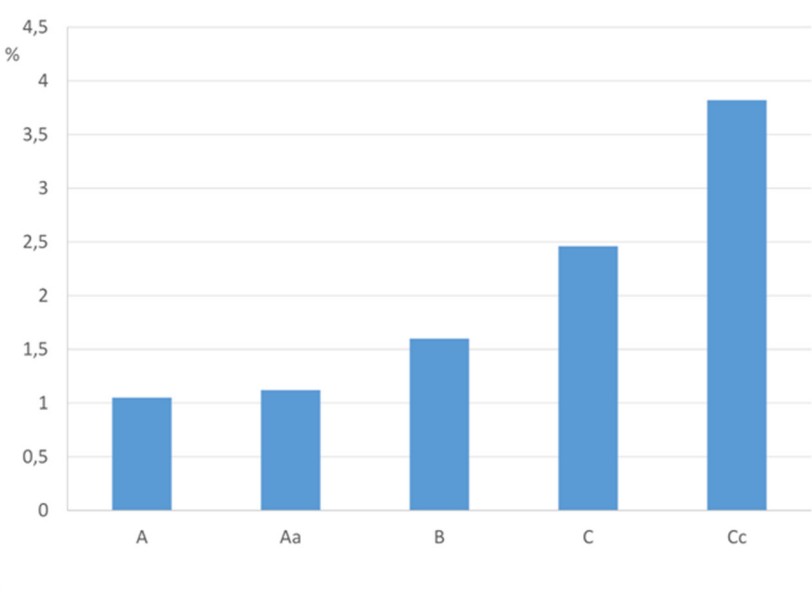

b

**Figure 8.** Variation of the concentration of some major elements (a) and of the LOI (loss on ignition) (b) in relation to the increase in in the degree of alteration (types A, Aa, B, C, and Cc).

Indeed, sodium, calcium, and potassium, mainly concentrated in feldspars, undergo a slight decrease linked to the decay of these minerals, especially plagioclase more attacked by the sericitic alteration.

On the contrary, magnesium remains constant as the alteration progresses, and this may seem anomalous, considering that in the unweathered rock, it is mainly contained in the biotite, which, as observed also in thin section, was markedly altered. However, the presence in the more altered facies of clay minerals such as chlorite and vermiculite allow us to reasonably hypothesise that, after its mobilisation, magnesium is fixed in the network of the aforementioned minerals.

The total Fe does not vary significantly; however, as is obvious, the decrease in FeO causes an increase in $Fe_2O_3$. Nerveless the oxidation of Fe may be a result from both

hydrothermal alteration and weathering. The LOI (loss on ignition) increases with the progress of the decay in a constant and sharp way to be a discriminating parameter to be used as an index of the decay (Figure 8b). To assess chemical weathering, many indices have been proposed and tested. These indexes consider a decrease in mobile cations (Ca, Na, K) and compare them with immobile ones (Al, Fe, Si). In Table 3, some selected chemical weathering indexes are reported [32–36], and in Table 4, they are calculated for the type A, Aa, B, C, and Cc samples.

**Table 3.** Chemical weathering indexes selected.

| Chemical Weathering Index | Key | Formula | Reference |
|---|---|---|---|
| Parker index | Wp | $[(2Na_2O/0,35)+(MgO/0,90)+(2K_2O/0,25)+(CaO/0,70)]*100$ | [32] |
| Chemical index of alteration | CIA | $[(Al_2O_3)/(Al_2O_3+ CaO+ Na_2O+ K_2O)]*100$ | [33] |
| Leaching coefficient | Lc | $SiO_2/(K_2O+Na_2O+CaO+MgO)$ | [34] |

**Table 4.** Results of the calculation of chemical weathering indexes.

| Chemical Weathering Index | A | Aa | B | C | Cc |
|---|---|---|---|---|---|
| Wp | 74.39 | 72.09 | 69.50 | 65.80 | 64.40 |
| CIA | 52.78 | 54.48 | 56.74 | 59.23 | 61.52 |
| Lc | 6.93 | 7.54 | 7.80 | 8.19 | 8.19 |

These indexes are in agreement with what is observed in Figure 8 regarding the alteration trend, i.e., leaching of mobile cations (K, Na, Ca) with their consequent decrease and relative increase in immobile phases (Al, Fe, Si) from samples A up to Cc.

### 4.4. Physical Characteristics

The average values of the physical characteristics of types A, Aa, and B are shown in Table 5. The alteration process determines a significant increase in the total open porosity (P%), which goes from 2.2% in the unweathered rock to 4.1% in the moderately altered one (Aa), up to 8.6% in the highest grade of alteration but still in a state of coherence (B).

**Table 5.** Physical characteristics type A, Aa, and B.

| | | P % | CIv % | IS% |
|---|---|---|---|---|
| A | M | 2.2 | 1.5 | 68 |
| | σ | 0.4 | 0.2 | 12 |
| Aa | M | 4.1 | 3.0 | 73 |
| | σ | 1.8 | 1.5 | 7 |
| B | M | 8.6 | 6.3 | 73 |
| | σ | 1.1 | 1.3 | 6 |

P % = total open porosity; CIv % = imbibition coefficient in volume; IS % = saturation index; M = average of 50 samples; σ = standard deviation.

The saturation index (IS%), with increasing alteration, remains substantially constant, i.e., classes of pores that retain more water are not formed, particularly those of capillary size.

Regarding the prevailing type of degradation, i.e., physical or chemical, the nature is intermediate. This is in agreement with the climatic conditions of the island, which is located in the middle of the Mediterranean environment where physical and chemical alteration are in a substantial equilibrium. Indeed, the thermal changes trigger thermoclastic phenomena with formation of a small intercrystalline porosity. This porosity, favouring the circulation of water and thanks to the relatively high

temperatures, determines the onset of hydrolysis phenomena (sericitisation of feldspars, chloritisation of biotite) with consequent leaching of Ca, Na, and K. However, this is a "weak" hydrolysis, since, with the progress of decay, a loss of coherence of the sound rock is essentially observed without major compositional variations occurring.

### 4.5. Biocolonisation

The phototrophic colonisation of the studied granite stones was mainly composed of different types of lichens (Figure 9) having various colours, such as *Acarospora fuscata*, *Candelariella vitellina*, and *Lecanora sulfurea* often associated with *Tephromela atra*, *Lecanora gangaleoides*, *Ochrolechia parella*, *Parmelia conspersa*, and *Parmelia loxodes*. Black patinas were also observed, mainly on vertical surfaces exposed to the north with a dominance of cyanobacteria belonging to Chroococccales (*Chroococcidiopsis* sp., *Gloeocapsa* sp.) and Nostocales (*Calothrix* sp., *Phormidium* sp.) groups. The biodeteriorarion of granite goes side by side with its weathering, being induced mainly by the crustose species, which have an intimate contact with the stone substrata and are able to cause not only mechanical damages but also chemical upsets.

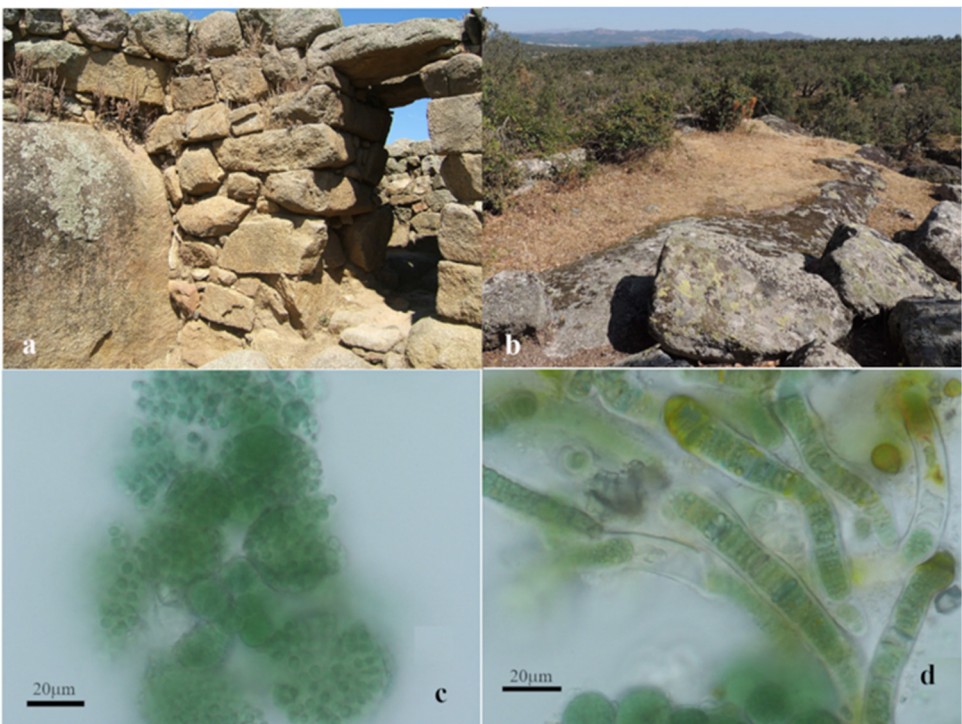

**Figure 9.** Biocolonisers of granite: (a) Parmelia conspersa (green greyish) and black patina; (b) Lecanora sulfurea (green yellowish), Tephormela atra (grey), and Parmelia loxodes (dark brown on the horizontal surface); (c) Chroococcidiopsis sp.-like colonies; (d) Calothrix sp.-like filaments.

### 5. Conclusions

The study of the alteration processes of Giglio granite has shown that already in the bedrock, there are small percentages of clay minerals such as kaolinite and chlorite due to hydrothermal alteration phenomena.

Starting from this condition, the exogenous alteration is initially expressed by physical phenomena (increase in intercrystalline porosity), which in turn favours the establishment of a slight hydrolysis of the silicates, as indicated in feldspars by their partial sericitisation and in biotite by chloritisation.

The chemical analysis showed a general decrease in Na, Ca, and K, in accordance with the chemical weathering indexes and the presence of a sericitisation process.



The alteration then proceeded with a further strong increase in the intercrystalline porosity up to the loss of coherence, which occurs when the latter exceeds 9%.

This increase in porosity determines an increase in the stone bioreceptivity as well. It is therefore relevant to highlight that a very high increase in porosity (an increase of about 300%) apparently does not correspond to a situation of such serious decay. This decay became evident immediately afterwards, with an almost sudden collapse of the structure.

Therefore, once again, the need for a continuous monitoring of the state of conservation of the architectural heritage is evident to avoid performing interventions when the situation is compromised.

As regards the chemical weathering indexes, these can be a support to objectively quantify the conservation conditions of the material. However, it must be considered that the actual conditions of conservation of a stone material in place also depend on the type of installation and the conditions of exposure to atmospheric agents and capillary rise phenomena. This can determine the fact that the material may have a low chemical weathering index, even though it is actually very decayed.

**Supplementary Materials:** The following supporting information can be downloaded at www.mdpi.com/article/10.3390/app12094588/s1.

**Author Contributions:** Please note that all authors of this paper should be considered as principal authors. The concept of the research has been shared by all authors. Sampling by F.F. and P.P.; biocolonisation study by O.A.C.; geological setting and geochemical analysis by P.P.; mineralogical and petrographic study by F.F.; petrophysical analysis by S.R. All the authors contributed to the discussion of the results and the drafting of the article. All authors have read and agreed to the published version of the manuscript.

**Funding:** This research received no external funding.

**Institutional Review Board Statement:** Not applicable.

**Informed Consent Statement:** Not applicable.

**Data Availability Statement:** Not applicable.

**Conflicts of Interest:** The authors declare that the research was conducted in the absence of any commercial or financial relationships that could be construed as a potential conflict of interest.

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
