# Peer review of "The Alteration of Giglio Island Granite: Relevance to the Conservation of Monumental Architecture"

_applsci, doi:10.3390/app12094588_

Round 1

Reviewer 1 Report

The authors present a detailed analysis of alteration phenomena of the Giglio island granite. They started with a wide historical and geological overview followed by petrographic, geochemical investigations. The dataset is enriched by the determination of some physical parameters, including the porosity, and the observation of the most common phototrophic colonizers. The work incorporates the knowledge derived from tens of years of detailed and fruitful studies on the decay of granitic rocks.

Overall, the manuscript is a very nice contribution and should be of interest to the readers of this journal.

I don't have substantive comments or criticisms regarding the author's methods, results or conclusions regarding the interpretation of the dataset.
On the other hand, here are some errors in the text with missing (or too much) spaces between words and sentences. I feel like it could be an issue with a pdf conversion. I trust that these will be easily fixed. As a minor note, English can be improved in places.

Author Response

We thank the reviewer for helpful suggestions and corrections to improve the scientific quality and understanding of the work. We have taken these into account and tried to integrate them appropriately into the text.

Reviewer 2 Report

Dear Editor

The manuscript by Fratini et al. contains data about the granite of Giglio Island. I have carefully read the manuscript and tried to critic. In my opinion, I think that the presentation of data is inadequate.  I am sorry but the present form of paper is not acceptable for publication. Accordingly, after mandatory major revisions, it can be published in your journal.

You can find my reviews on the article in the attached file.

With my best wishes

Author Response

Reply to revisor 2

We thank the reviewer for helpful suggestions and corrections to improve the scientific quality and understanding of the work. We have taken these into account and tried to integrate them appropriately into the text.

The title is too general and does not reflect the present study. It is recommended to replace.

Done

It would be nice if a scale could be put on picture 2.

We don’t think to add a scale to the picture because the dimension is clear: there is the entrance door of the bell tower which give the idea of the dimension

Figure 3 and Figure 4 can be combined as a single picture.

Done

It would be nice to use color instead of hue.

Done

Figure 5, 6 and Figure 7 can be combined as a single picture.

Done

This section is a bit complex and not well organized. It would be appropriate to rewrite it.

Firstly, the general texture of the rock, its mineral composition, then the characteristics of the minerals and their weathering conditions should be given.

What is the meaning of Pecilitic?

Removed

maybe mafic!!!

Corrected

In this section, it would be good if you put the thin-section photographs of the samples that show different alteration degree.

We have not realized all the thin sections in the different degree of alteration

Figure 11 and Figure 12 can be combined as a single picture.

Done

In this section, XRD graphs of granites showing different weathering should be given. However, XRD graphics of decomposition products (clay minerals) should be included.

We did not do XRDs of the main mineral composition of the granite samples because we wanted to study the clay minerals. In order to correctly identify the clay minerals the procedure is to realize 4 spectra (untreated, glycolate , fired at 450 and 600 °C).  The comparison of these spectra allows the correct identification of the composition of the clay minerals. The spectra are acquired in hard copy in order to compare them with each other. There is no single spectrum to be presented in order to show the variation of mineral composition from sound to altered material. Inserting all the spectra would make the diagrams illegible.

This table is nice, but it would be appropriate to give it as % values by determining it from XRD graphics.

In the past we used to insert percentages by making a relative calculation on the graphs, but we have been repeatedly challenged that only the use of the Rietveld method allows to indicate percentages. Therefore, we decided not to put percentages anymore. Our method is a semi-quantitative analysis and as such we indicate the relative abundances in this way.

Diagrams that show the composition of the rock according to the major oxide element contents should be drawn (it is recommended to see Koralay and Çelik 2019 on this subject).

We cannot figure out which diagrams you are referring to. Keep in mind that we had to remove the diagrams Strekeisen QAPF and A'KF in order to accomplish the strange request of the Assistant Editor to provide the copyright license certificate. We never had a similar request for a modified diagram

Weathering indices have been suggested to describe the intensity of weathering depending upon the nature and requirement of the study. In this section, some weathering indices should be calculated and the weathering status of the rocks should be revealed more clearly (see Koralay and Çelik 2019 and Deniz et al. 2021 on this subject).

Some indexes have been inserted

As the weathering degree increases, the K2O value is expected to increase. On your chart, it is decreasing on the contrary. What is your explanation for this situation?

As reported by Parker, 1970; Irfan, 1996; Gupta and Rao, 2001; Ng et al., 2001; Arel and Tugrul, 2001; Kim and Park, 2003 et others researcher weathering indices are based on the decrease of some mobile cations (Ca, Na, K) compared with immobile ones (Al, Fe, Si), expressed as ratios Therefore, the decrease of K2O, that have measured in our samples, is in agreement what reported by scholars

There is no significant increase and/or decrease in TiO2 and MgO values, so it would be better to exclude them from the chart. In addition, the explanations for TiO2, which is considered to be stable during alteration and metamorphism events, were not found realistic.

Done

please put (+) instead of 1:

We had to remove the diagrams A'KF in order to accomplish the strange request of the Assistant Editor to provide the copyright license certificate

Line 310- 323, This section is a bit complex and not well organized. It would be appropriate to rewrite it.

Done

Conclusion This section is a bit complex. The conclusion does not fully cover the article findings. It should be rewritten according to the new additions to be made.

Done

Reference Many local references (Italian) were used. For an international reader, it is recommended to use new and current references.

Done

Reviewer 3 Report

The manuscript is interesting and I think it may be published in Applied Sciences after some important review. Congrats!  I add several comments in the following that I hope would help authors to optimize it. I am not native thus I apologize for some English mistake.

Main comments

May first main comment concerns to the discrimination between hydrothermal alteration and weathering, which appears to be not much clear. Hydrothermal (deuteric) alteration is intrinsic to the post-magmatic evolution of granitic rocks and may be heterogeneously distributed within a given pluton. It includes several processes as the substitution of biotite for chlorite, sericitization of plagioclase feldspars, penitization of cordierite, etc.  Weathering occurs much late and is a process affecting outcrops near or at the surface, thus it is not clear what are the hydrothermal alteration effects and what are the weathering effects and how each one contribute to the decay of the studied rocks. Of course hydrothermal effects may facilitate weathering. I understood that weathering decay is the main manuscript´s objective and the hydrothermal effects were already there when the rocks were quarried.

The second is related to the geologic and petrographic nomenclature. I understand that the area of expertise of the authors is conservation of building materials in the architectural heritage. However, several terms used in the main text and figures appear to be not appropriate in describing the geologic and petrographic properties of the rocks and should be adjusted. I may suggest some help from a specialist in the area.

Specific comments

Key words: granite weathering may not be too appropriate, because you have also studied hydrothermal alteration and it is very important.

Lines 102-106: I think “granodiorite and quartz-monzonite intrusive bodies have emplaced” is better. In the geochronological literature, usually Ma is used for million years.

Line 108: anatexis rather than anatexes

Lines 107-1019: You are using here inappropriate terms for describing sedimentary and metamorphic (metasedimentary) rocks. In the geological literature, argillite, quartz conglomerate and limestone are sedimentary rocks whereas schist and quartzite are metamorphic (metasedimentary) rocks. Also the granites were probably derived from melting of metasedimentary rather than sedimentary rocks.

Lines 120-123: This paragraph appears to be out of context

Lines 128-133: Monzogranites are rocks that plot within the monzogranite field in the QAP diagram (see your Figure 10), given by a range rather than absolute values of alkali-feldspar (A), plagioclase (P) and quartz (Q). Secondary is normally used for minerals which were derived by post-magmatic processes (hydrothermal alteration, weathering, etc.) and not for primary phases precipitated from the melt. Thus, biotite, cordierite, zircon, apatite, etc. are probably primary phases; titanite is probably secondary as it is not expected as a primary accessory in peraluminous rocks as the studied ones.

Lines 135-138: Petrographic analysis does not allow recognizing different magmatic intrusions per se. Note that one single intrusion may present significant petrographic varieties, originated by in situ fractional crystallization for instance. Here you need some geological and/or structural information.

Lines 139-146: Lineation and homogeneous as applied here are structural rather than textural rock properties. Texture refers to the arrangement of rock mineral constituents. Massive is better than homogeneous, I think. Note you may have and “homogeneously lineated” rock. I think megacryst is better than phenocryst as it is just a descriptive term. “auto-intrusions of xenoliths” is not appropriate. What you mean? Xenoliths are country rock fragments captured during the emplacement of the intrusion. There are some related terms in the literature as “autoliths”, “enclaves”, “micro-granular enclaves”, etc., each with their proper meaning.

Line 161: Again, I think massive is better than homogeneous;

Line 172: Methods is better than methodologies;

Line 181-184: Please add a phrase stating how you determine the plagioclase anorthite content (An content) under de microscope, as you mention latter that it has an andesine-oligoclase composition. The QAPF diagram includes the lower triangle APF which applies for silica under-saturated rocks (feldspathoid-bearing rocks) which is not your case. Use QAP.

Line 185: X ray or X-ray better than Xray;

Line 193: Conventional major and minor elements rather than solely major elements. Note that  Mn is usually taken as a minor element in rocks such as the studied ones.

Lines 196-203: here and in all this section, you may be more objective and to the point. On the other hand it would be nice if you define CI and IS or cite a reference.

Line 2011: Monzogranite is a plutonic rock hence you do not need to state that it has a holocrystalline texture.

Lines 212-237. I may suggest you search for some petrographic description of igneous rocks presentation in the literature in order to optimize your descriptions. Crystal cores and rims are better than crystal centers and edges; perhaps “anhedral” would be better than “alotriomorphic” , note you use euhedral before;  I am not sure the term “pelicitic” applies. Femic is for Fe-Mg minerals and cordierite is also femic. In general I may suggest the use of felsic (feldspars, quartz) and mafic (biotite, cordierite, tourmaline, accessory phases, etc.) minerals.

I may suggest adding a Table (may be a supplementary table) with the modal results.

Lines 268-272: The mineral assemblage (biotite + cordierite + tourmaline) indicates that the studied rocks are peraluminous. Here you may add the Alumina Saturation Index (ASI = Al2O3/(CaO+Na2O+K2O) in molecular proportions) to show it is greater than 1, as expected in peraluminous rocks, which is compatible with a metasedimentary source for the original melts (see Chappel & White). I am not sure “class” applies here. The currently classification of plutonic rocks according the IUGS is based on their modal contents, thus you need not “confirm” this with geochemical information.

Lines 273-291: may b simplified.  This may be a hard task but in some section it would be interesting if you contrast the products of hydrothermal or deuteric alteration (formation of chlorite, sericite and penine, etc.) from the more typical weathering products (perhaps kaolinite, etc.). See main comments

Lines 286-287: oxidation may be a result from both hydrothermal alteration and weathering;

Line 307: a symbol for the standard deviation is missing;

Figures.

In general I think that some figure captions should be extended and more explicative.

Figure 1. Location rather than position

Figure 2. Need it be so large?

Figure 3. I am not sure blocks are an adequate term here;

Figure 5. The figure is showing some particular structure (clear veins). Please explain

Figure 6. I am not sure that the term inclusion applies here. State that the rock has a massive structure.

Figure 6. The figure shows some structures within the granite. Are they xenoliths? Syn-plutonic dikes? Explain.

Figure 10. This must be an equilateral triangle. Please solely  QAP in the caption.

Figure 11. Need not to put 100x, as a graphic scale is given. Crossed polarizers is better than nicol ^. The discolored aspect is not due to Fe-hydroxides concentration. It is probably due to hydrothermal alteration and formation of secondary minerals. I think it is chlorite rather than penine.

Figure 13. Perhaps it would be more illustrative if you plot oxide ratios (normalized to the less altered sample, in an appropriate scale) vs. samples. Please adjust number subscripts in the oxides (e.g., SiO2 rather than SiO2). Concentration is given in wt.% oxides and LOI.

Figure 14. I think the observed trend is also (at least in part) related to hydrothermal alteration and not solely from weathering as stated in the legend. See main comments.

Figure 15. Put and/or optimize the scales. It appears there are not scales for figures (a) and (b) and they are not legible in (c) and (d).

Line360-367: Please, adjust.

Tables

I may suggest adding a supplementary table with the modal results, emphasizing the proportions of the alteration minerals.

Table 2: Please add the totals in each result. The sigma symbol is missing in the table footnote. What you mean for “di”?

Author Response

Reply to revisor 3

We thank the reviewer for helpful suggestions and corrections to improve the scientific quality and understanding of the work. We have taken these into account and tried to integrate them appropriately into the text.

The manuscript is interesting and I think it may be published in Applied Sciences after some important review. Congrats!  I add several comments in the following that I hope would help authors to optimize it. I am not native thus I apologize for some English mistake.

Main comments

May first main comment concerns to the discrimination between hydrothermal alteration and weathering, which appears to be not much clear. Hydrothermal (deuteric) alteration is intrinsic to the post-magmatic evolution of granitic rocks and may be heterogeneously distributed within a given pluton. It includes several processes as the substitution of biotite for chlorite, sericitization of plagioclase feldspars, penitization of cordierite, etc.  Weathering occurs much late and is a process affecting outcrops near or at the surface, thus it is not clear what are the hydrothermal alteration effects and what are the weathering effects and how each one contribute to the decay of the studied rocks. Of course hydrothermal effects may facilitate weathering. I understood that weathering decay is the main manuscript´s objective and the hydrothermal effects were already there when the rocks were quarried.

Done

The second is related to the geologic and petrographic nomenclature. I understand that the area of expertise of the authors is conservation of building materials in the architectural heritage. However, several terms used in the main text and figures appear to be not appropriate in describing the geologic and petrographic properties of the rocks and should be adjusted. I may suggest some help from a specialist in the area.

Thank you for your suggestions to improve the text and your understanding of the work

Specific comments

Key words: granite weathering may not be too appropriate, because you have also studied hydrothermal alteration and it is very important.

 Done

Lines 102-106: I think “granodiorite and quartz-monzonite intrusive bodies have emplaced” is better. In the geochronological literature, usually Ma is used for million years.

Done

Line 108: anatexis rather than anatexes.

Done

Lines 107-1019: You are using here inappropriate terms for describing sedimentary and metamorphic (metasedimentary) rocks. In the geological literature, argillite, quartz conglomerate and limestone are sedimentary rocks whereas schist and quartzite are metamorphic (metasedimentary) rocks. Also the granites were probably derived from melting of metasedimentary rather than sedimentary rocks.

Done

Lines 120-123: This paragraph appears to be out of context.

Done

Lines 128-133: Monzogranites are rocks that plot within the monzogranite field in the QAP diagram (see your Figure 10), given by a range rather than absolute values of alkali-feldspar (A), plagioclase (P) and quartz (Q). Secondary is normally used for minerals which were derived by post-magmatic processes (hydrothermal alteration, weathering, etc.) and not for primary phases precipitated from the melt. Thus, biotite, cordierite, zircon, apatite, etc. are probably primary phases; titanite is probably secondary as it is not expected as a primary accessory in peraluminous rocks as the studied ones.

Done

Lines 135-138: Petrographic analysis does not allow recognizing different magmatic intrusions per se. Note that one single intrusion may present significant petrographic varieties, originated by in situ fractional crystallization for instance. Here you need some geological and/or structural information.

Done

Lines 139-146: Lineation and homogeneous as applied here are structural rather than textural rock properties. Texture refers to the arrangement of rock mineral constituents. Massive is better than homogeneous, I think. Note you may have and “homogeneously lineated” rock. I think megacryst is better than phenocryst as it is just a descriptive term. “auto-intrusions of xenoliths” is not appropriate. What you mean? Xenoliths are country rock fragments captured during the emplacement of the intrusion. There are some related terms in the literature as “autoliths”, “enclaves”, “micro-granular enclaves”, etc., each with their proper meaning.

Done

Line 161: Again, I think massive is better than homogeneous;

Done

Line 172: Methods is better than methodologies;

Done

Line 181-184: Please add a phrase stating how you determine the plagioclase anorthite content (An content) under de microscope, as you mention latter that it has an andesine-oligoclase composition. The QAPF diagram includes the lower triangle APF which applies for silica under-saturated rocks (feldspathoid-bearing rocks) which is not your case. Use QAP.

We had to remove the diagrams Strekeisen QAPF and A'KF in order to accomplish the strange request of the Assistant Editor to provide the copyright license certificate. We never had a similar request for a modified diagram

Line 185: X ray or X-ray better than Xray;

Done

Line 193: Conventional major and minor elements rather than solely major elements. Note that  Mn is usually taken as a minor element in rocks such as the studied ones.

Done

Lines 196-203: here and in all this section, you may be more objective and to the point. On the other hand it would be nice if you define CI and IS or cite a reference.

Done

Line 2011: Monzogranite is a plutonic rock hence you do not need to state that it has a holocrystalline texture.

Done

Lines 212-237. I may suggest you search for some petrographic description of igneous rocks presentation in the literature in order to optimize your descriptions. Crystal cores and rims are better than crystal centers and edges; perhaps “anhedral” would be better than “alotriomorphic” , note you use euhedral before;  I am not sure the term “pelicitic” applies. Femic is for Fe-Mg minerals and cordierite is also femic. In general I may suggest the use of felsic (feldspars, quartz) and mafic (biotite, cordierite, tourmaline, accessory phases, etc.) minerals.

Done

I may suggest adding a Table (may be a supplementary table) with the modal results

Modal analysis was carried out only on type A samples, i.e. the macroscopically unweathered granite, in order to classify the rock.

Lines 268-272: The mineral assemblage (biotite + cordierite + tourmaline) indicates that the studied rocks are peraluminous. Here you may add the Alumina Saturation Index (ASI = Al2O3/(CaO+Na2O+K2O) in molecular proportions) to show it is greater than 1, as expected in peraluminous rocks, which is compatible with a metasedimentary source for the original melts (see Chappel & White). I am not sure “class” applies here. The currently classification of plutonic rocks according the IUGS is based on their modal contents, thus you need not “confirm” this with geochemical information.

Done

Lines 273-291: may b simplified.  This may be a hard task but in some section it would be interesting if you contrast the products of hydrothermal or deuteric alteration (formation of chlorite, sericite and penine, etc.) from the more typical weathering products (perhaps kaolinite, etc.). See main comments

Done

Lines 286-287: oxidation may be a result from both hydrothermal alteration and weathering;

Done

Line 307: a symbol for the standard deviation is missing;

Done

Figures.

In general I think that some figure captions should be extended and more explicative.

Figure 1. Location rather than position

Done

Figure 2. Need it be so large?

Done

Figure 3. I am not sure blocks are an adequate term here;

Done

Figure 5. The figure is showing some particular structure (clear veins). Please explain.

Done

Figure 6. I am not sure that the term inclusion applies here. State that the rock has a massive structure.

Done

Figure 10. This must be an equilateral triangle. Please solely  QAP in the caption.

We had to remove the diagrams Strekeisen QAPF and A'KF in order to accomplish the strange request of the Assistant Editor to provide the copyright license certificate. We never had a similar request for a modified diagram

Figure 11. Need not to put 100x, as a graphic scale is given. Crossed polarizers is better than nicol ^. The discolored aspect is not due to Fe-hydroxides concentration. It is probably due to hydrothermal alteration and formation of secondary minerals. I think it is chlorite rather than pennine.

Done

Figure 13. Perhaps it would be more illustrative if you plot oxide ratios (normalized to the less altered sample, in an appropriate scale) vs. samples. Please adjust number subscripts in the oxides (e.g., SiO2 rather than SiO2). Concentration is given in wt.% oxides and LOI.

Done

Figure 14. I think the observed trend is also (at least in part) related to hydrothermal alteration and not solely from weathering as stated in the legend. See main comments.

We had to remove the diagrams Strekeisen QAPF and A'KF in order to accomplish the strange request of the Assistant Editor to provide the copyright license certificate. We never had a similar request for a modified diagram

Figure 15. Put and/or optimize the scales. It appears there are not scales for figures (a) and (b) and they are not legible in (c) and (d).

Done

Line360-367: Please, adjust.

Done

Tables

I may suggest adding a supplementary table with the modal results, emphasizing the proportions of the alteration minerals.

Modal analysis was carried out only on type A samples, i.e. the macroscopically unweathered granite, in order to classify the rock.

Table 2: Please add the totals in each result. The sigma symbol is missing in the table footnote. What you mean for “di”?

Done

Round 2

Reviewer 2 Report

I would like to state that I do not find the answers (except for a few) very satisfactory by the authors to the issues that I would like to be edited.

I would like to put a scale on picture 2.

The authors give the answer "We don’t think to add a scale to the picture because the dimension is clear: there is the entrance door of the bell tower which gives the idea of the dimension"

That is ok but I am not living in Italy and I don't know how high the entrance door is.

Another one, I suggest that it would be good if you put the thin-section photographs of the samples that show different alteration degrees.

The authors gives "We have not realized all the thin sections in the different degrees of alteration"

You will be working on a subject such as alteration/weathering and you will not notice the alteration/weathering products in thin sections,  It is not seen as an acceptable answer.

Another recommendation "In this section, XRD graphs of granites showing different weathering should be given. However, XRD graphics of decomposition products (clay minerals) should be included.

The authors give the answer as "We did not do XRDs of the main mineral composition of the granite samples because we wanted to study the clay minerals. In order to correctly identify the clay minerals the procedure is to realize 4 spectra (untreated, glycolate , fired at 450 and 600 °C).  The comparison of these spectra allows the correct identification of the composition of the clay minerals. The spectra are acquired in hard copy in order to compare them with each other. There is no single spectrum to be presented in order to show the variation of mineral composition from sound to altered material. Inserting all the spectra would make the diagrams illegible."

While such table representations mean many things for social sciences, they mean nothing but presence/absence for engineering and geoscientists. If they studied clay minerals in granite why did not the authors put the clay mineral XRD graphs? 

Another one "As the weathering degree increases, the K2O value is expected to increase. On your chart, it is decreasing on the contrary. What is your explanation for this situation?"

The authors give the answer as "As reported by Parker, 1970; Irfan, 1996; Gupta and Rao, 2001; Ng et al., 2001; Arel and Tugrul, 2001; Kim and Park, 2003 et others researcher weathering indices are based on the decrease of some mobile cations (Ca, Na, K) compared with immobile ones (Al, Fe, Si), expressed as ratios, Therefore, the decrease of K2O, that have measured in our samples, is in agreement what reported by scholars"

Is that all? Is there any logical explanation? Can't this be associated with the mineralogical composition?

Author Response

Reply to Revisor 2, 2nd round

Comments and Suggestions for Authors

I would like to state that I do not find the answers (except for a few) very satisfactory by the authors to the issues that I would like to be edited.

I would like to put a scale on picture 2. The authors give the answer "We don’t think to add a scale to the picture because the dimension is clear: there is the entrance door of the bell tower which gives the idea of the dimension". That is ok but I am not living in Italy and I don't know how high the entrance door is.

The  scale has been put in the figure

Another one, I suggest that it would be good if you put the thin-section photographs of the samples that show different alteration degrees.

The authors gives "We have not realized all the thin sections in the different degrees of alteration"

You will be working on a subject such as alteration/weathering and you will not notice the alteration/weathering products in thin sections,  It is not seen as an acceptable answer.

We apologise but we did not understand your request correctly. The sections of samples A Aa B C have been made otherwise we would not have been able to do the petrographic study. The section of type Cc was not made because the material is almost incoherent.

Images of thin sections of types A, Aa, B and C have been included as supplementary material

Another recommendation "In this section, XRD graphs of granites showing different weathering should be given. However, XRD graphics of decomposition products (clay minerals) should be included. The authors give the answer as "We did not do XRDs of the main mineral composition of the granite samples because we wanted to study the clay minerals. In order to correctly identify the clay minerals the procedure is to realize 4 spectra (untreated, glycolate , fired at 450 and 600 °C).  The comparison of these spectra allows the correct identification of the composition of the clay minerals. The spectra are acquired in hard copy in order to compare them with each other. There is no single spectrum to be presented in order to show the variation of mineral composition from sound to altered material. Inserting all the spectra would make the diagrams illegible." While such table representations mean many things for social sciences, they mean nothing but presence/absence for engineering and geoscientists. If they studied clay minerals in granite why did not the authors put the clay mineral XRD graphs? 

XRD diagrams of clay minerals of types A, Aa, B, C and Cc have been included as supplementary material.

Another one "As the weathering degree increases, the K2O value is expected to increase. On your chart, it is decreasing on the contrary. What is your explanation for this situation?"

The authors give the answer as "As reported by Parker, 1970; Irfan, 1996; Gupta and Rao, 2001; Ng et al., 2001; Arel and Tugrul, 2001; Kim and Park, 2003 et others researcher weathering indices are based on the decrease of some mobile cations (Ca, Na, K) compared with immobile ones (Al, Fe, Si), expressed as ratios, Therefore, the decrease of K2O, that have measured in our samples, is in agreement what reported by scholars"

Is that all? Is there any logical explanation? Can't this be associated with the mineralogical composition?

K is a mobile cation and, as reported by several authors, this element tends to decrease during exogenous alteration. The extent of the decrease depends on the climatic conditions at the site. In tropical climates the mobilisation of this element is very high, in cold climates it is less intense. In the case of the island of Giglio the climatic conditions are intermediate. As can be seen in Table 2, K, except in type Aa (slight increase), shows a slight decrease (types B, C, Cc) with respect to type A. This is in agreement with what is reported by the majority of scholars on which they also calculate the index of chemical alteration (Parker, 1970; Irfan, 1996; Gupta and Rao, 2001; Ng et al., 2001; Arel and Tugrul, 2001; Kim and Park, 2003)

Reviewer 3 Report

A revised text was prepared however I think you need to rethink and provide a more careful and integrated revision. I am suggesting moderate revisions as in may opinion you have a significant amount of work to have a good paper! I will mention a few aspects for instance:

I am not sure if the title is good, as the main point of the manuscript appears to be related to the conservation of architectural heritage .

There is significant background on granite alteration and implications for the conservation of architectural heritage. No specific mentions/references in the introduction section!

Several among by previous comments were considered, however I think they should be better considered. For instance:

phyllite is a foliated metamorphic rock (low grade), not a sedimentary rock

quartzite veins? quartzite is a metamorphic rock....perhaps quartz veins...

How was the anorthite content IN PLAGIOCLASE determined under the microscope? Here you should present the used method!

Crossed polarizers nicols????? nicols are calcite combined crystals used as polarizers in older optical microscopes. Just crossed polarizers.

Of course the ASI parameter is a primary parameter and thus should be used for unaltered or just slightly altered rocks. As it is expected it increases also with weathering  intensity, as demonstrated by the CIA index.

I think the CIA formula is not correct. I did not check the others

Author Response

Reply to Revisor 3, 2nd round

Comments and Suggestions for Authors

A revised text was prepared however I think you need to rethink and provide a more careful and integrated revision. I am suggesting moderate revisions as in may opinion you have a significant amount of work to have a good paper! I will mention a few aspects for instance:

I am not sure if the title is good, as the main point of the manuscript appears to be related to the conservation of architectural heritage.

The title has been modified as in the following:

“The alteration of Giglio island granite: relevance to the conservation of monumental architecture”

There is significant background on granite alteration and implications for the conservation of architectural heritage. No specific mentions/references in the introduction section!

The following references have been added:

Matias J. M. S., Alves C. A. S. The influence of petrographic, architectural and environmental factors in decay patterns and durability of granite stones in Braga monuments (NW Portugal). 2002, Geological Society, London, Special Publications, 205: 273-281

Lee C.H., Lee M.S., Suh M., Choi S.W.. Weathering and deterioration of rock properties of the Dabotap pagoda (World Cultural Heritage), Republic of Korea. 2005, Environ Geol 47: 547–557

Tomás R., Cano M., Pulgarín L.F., Brotóns V., Benavente D., Miranda T., Vasconcelos G..,Thermal effect of high temperatures on the physical and mechanical properties of a granite used in UNESCO World Heritage sites in north Portugal. 2021, Journal of Building Engineering, 43:102823

Navarro R., Monterrubio S., Pereira D. The Importance of Preserving Small Heritage Sites: the Case of La Tuiza Sanctuary (Zamora, Spain). 2022, Geoheritage 14, 47

Storta E., Borghi A., Perotti L., Palomba M., Deodato A., Minero-petrographic characterization of stone materials used for the roman amphitheater of Eporedia (Ivrea, To): A scientific-dissemination proposal in the Cultural Heritage. 2022, Resources Policy, 77: 102668

Several among by previous comments were considered, however I think they should be better considered. For instance:

phyllite is a foliated metamorphic rock (low grade), not a sedimentary rock.

Yes, this is true. Therefore, we have corrected the sentence as follows in order not to create ambiguity:

“The intrusion of the monzogranite mass took place inside sedimentary rocks which  remnants from contact metamorphism can be observed at Punta del Fenaio (north of the island) in two small outcrops of strongly foliated phyllites”.

quartzite veins? quartzite is a metamorphic rock....perhaps quartz veins...

We have corrected the error

How was the anorthite content IN PLAGIOCLASE determined under the microscope? Here you should present the used method!

The Michel-Lévy method, on thin sections observed at the polarising microscope, has been used according to what reported in (Nesse, 2000). The method utilises a minerals’ known extinction angles and maximum birefringence to indicate the abundances of Na (Albite) and Ca (Anorthite) in sub- to euhedral twinned plagioclase, using thin sections and a polarising microscope

Crossed polarizers nicols????? nicols are calcite combined crystals used as polarizers in older optical microscopes. Just crossed polarizers.

Done

Of course the ASI parameter is a primary parameter and thus should be used for unaltered or just slightly altered rocks. As it is expected it increases also with weathering  intensity, as demonstrated by the CIA index.

We have corrected by leaving ASI index only for type A

I think the CIA formula is not correct. I did not check the others

We have corrected the CIA index formula
